# Hollow Bioelectrodes Based on Buckypaper Assembly. Application to the Electroenzymatic Reduction of O_2_

**DOI:** 10.3390/nano12142399

**Published:** 2022-07-14

**Authors:** Paulo Henrique M. Buzzetti, Anastasiia Berezovska, Yannig Nedellec, Serge Cosnier

**Affiliations:** University Grenoble Alpes, CNRS, DCM, 38000 Grenoble, France; paulo-henrique.maciel-buzzetti@univ-grenoble-alpes.fr (P.H.M.B.); anastasiia.berezovska@univ-grenoble-alpes.fr (A.B.); yannig.nedellec@univ-grenoble-alpes.fr (Y.N.)

**Keywords:** hollow bioelectrode, buckypaper, microcavity, oxygen electroreduction, bilirubin oxidase

## Abstract

A new concept of hollow electrode based on the assembly of two buckypapers creating a microcavity which contains a biocatalyst is described. To illustrate this innovative concept, hollow bioelectrodes containing 0.16–4 mg bilirubin oxidase in a microcavity were fabricated and applied to electroenzymatic reduction of O_2_ in aqueous solution. For hemin-modified buckypaper, the bioelectrode shows a direct electron transfer between multi-walled carbon nanotubes and bilirubin oxidase with an onset potential of 0.77 V vs. RHE. The hollow bioelectrodes showed good storage stability in solution with an electroenzymatic activity of 30 and 11% of its initial activity after 3 and 6 months, respectively. The co-entrapment of bilirubin oxidase and 2,2-azino-bis(3-ethylbenzothiazoline-6-sulfonic acid) in the microcavity leads to a bioelectrode exhibiting mediated electron transfer. After 23 h of intermittent operation, 5.66 × 10^−4^ mol of O_2_ were electroreduced (turnover number of 19,245), the loss of catalytic current being only 54% after 7 days.

## 1. Introduction

Enzymatic biofuel cells that convert chemical energy into electrical energy by electroenzymatic reactions offer attractive potentialities for powering disposable electronic systems [1,2]. However, the development of enzymatic biofuel cells is confronted with two major technological obstacles, namely their short operational and storage lifetime in solution and, to a lesser extent, their low power output [3,4]. The low stability of the bioelectrodes of the enzymatic fuel cells is linked to the deactivation of the immobilized enzymes and seems inevitable. Regarding the power of the biofuel cells, the latter is partly related to the quantity of enzyme immobilized per unit of conductive surface and to the efficiency of the electrical connection between the enzyme and the electrode.

The enzyme electrodes of biofuel cells result from the immobilization of different redox enzymes on the surfaces of the electrodes for their electrical connection. This fixation of enzyme can be obtained by chemical grafting or affinity interactions or by physical entrapment. Regarding the immobilization by covalent or non-covalent bond, this configuration offers good access of the substrate and redox mediators to the immobilized enzyme, but the quantity of biocatalyst is limited to a quasi-monolayer at the modified electrode–solution interface, thus strongly limiting the power.

Although immobilization in a 3D matrix increases the surface density of enzymes, this entrapment process induces a denaturation process due to the non-biocompatible environment. Additionally, the activity of the entrapped enzyme can be affected by the permeability and hydrophobicity of the host structure. Steric constraints can drastically reduce the permeation of substrates and redox mediators or even block the conformational flexibility of the protein. One way to increase the power of biofuel cells is to develop porous electrodes that increase the effective surface area of the electrodes. However, this approach does not solve the storage stability problem.

Recently, a new type of biofuel cell based on the non-immobilization of catalysts has been developed based on compartments comprising a dialysis membrane and containing enzymes, redox nanoparticles and an electrode [5]. As enzymes and mediators in solution can freely diffuse and rotate, this approach allows good orientation favoring electrical wiring of the enzymes. However, this process requires a peristaltic pump to provide a constant flow of fuel and hence is difficult to miniaturize. More recently, Li et al. reported a novel design of a bioelectrode based on a carbon felt electrode and an aqueous slurry of glucose oxidase, electron mediator and dispersed nanomaterial (graphene-like Ti_3_C_2_ MXene) confined in an acrylic shell and a dialysis membrane [6]. The maximum current density of this bioanode is maintained at 25% of its initial value after 9 days, illustrating the relatively good storage stability of the bioanode. In addition, for a low continuous current discharge (5 µA), the resulting hybrid biofuel cell exhibits good operational stability, losing only 34% of its initial power (2.75 µW) after 19 days. However, the biofuel cell configuration based on an acrylic container wrapped in a dialysis membrane and the stability of the dispersion of the nanomaterial can be limiting factors for the miniaturization of the biofuel cell and its stability.

Recently, we have demonstrated that a deposit of a thin layer of carbon nanotube (CNT) on a non-conductive support could play the role of an electrode while offering good permeation to low molecular weight compounds in an aqueous medium [7]. Taking into account this property, we report here the original creation of a hollow planar bioelectrode of very low thickness and large surface area containing the enzyme in powder form. The enzyme is trapped during the bonding of two conductive sheets made up of carbon nanotubes (buckypaper), the volume of the microcavity being defined by the thickness of the glue film binding these buckypapers. The buckypapers being permeable to water and small molecules but not allowing the permeation of enzymes, the bioelectrode presents a high density of protein in a microvolume. Moreover, the bioelectrode can be stored dry and the enzyme is only solubilized during the use of the bioelectrode.

To illustrate this innovative concept, a hollow bioelectrode configuration with entrapped bilirubin oxidase (BOx) was fabricated and applied to electroenzymatic reduction of O_2_. The electrocatalytic performance of the BOx bioelectrode was described as a function of pH, temperature and the amount of entrapped enzyme. The operational and storage stability of the bioelectrode in solution have been determined via the evolution of the catalytic current.

In addition, the influence of the iron-protoporphyrin (hemin) adsorbed on buckypaper on the orientation of BOx and therefore on the direct electron transfer (DET) was studied. Moreover, the effect of adding 2,2-azino-bis(3-ethylbenzothiazoline-6-sulfonic acid) diammonium salt (ABTS) as a redox mediator for mediated electron transfer (MET) on the electrical wiring of the BOx was also investigated.

## 2. Materials and Methods

### 2.1. Materials and Reagents

Sodium phosphate dibasic (Na_2_HPO_4_) and sodium phosphate monobasic (NaH_2_PO_4_) for the phosphate buffers preparation were purchased from Sigma-Aldrich (Paris, France). Multi-walled carbon nanotubes (MWCNTs; Ø = 9.5 nm, purity > 95%, 1.5 μm length) were obtained from Nanocyl and used for laboratory-made buckypaper (L_bp_) fabrication. Commercial MWCNT proprietary blend buckypaper (C_bp_) was obtained from Nano-TechLabs, Inc (reference number NTL-12218). N,N-dimethylformamide (DMF, 99.9%), iron-protoporphyrin (hemin; ≥97%) and 2,2-azino-bis(3-ethylbenzothiazoline-6-sulfonic acid) diammonium salt (ABTS, ≥98%) were purchased from Sigma-Aldrich. Bilirubin oxidase (BOx) from *Myrothecium verrucaria* (E.C. 1.3.3.5; 1.96 U mg^−1^) was purchased from Amano Enzyme. Oxygen and argon were purchased from Air Liquide (Paris, France).

### 2.2. Laboratory-Made Buckypaper Preparation and Characterization

Based on a vacuum filtration method previously reported by our research group, two different compositions were used for laboratory-made buckypaper fabrication [8].

Laboratory-made buckypaper (L_bp_): 66 mg of MWCNTs (1 mg mL^−1^) was added in 66 mL of DMF followed by sonication using a Bandelin Sonorex RK100 (Berlin, Germany) ultrasonic bath for 90 min. The resulting suspension was then filtrated using a diaphragm pump (MZ 2C NT model, Vaccubrand) on a Millipore PTFE filter (JHWP, 0.45 μm pore size). The L_bp_ was rinsed with water and left under vacuum for 1 h. Finally, the L_bp_ was removed from the filter and then left to dry in air overnight.Laboratory-made buckypaper modified by (hemin-L_bp_): 25 mg of hemin (0.6 mM) was added in 66 mL of DMF with 66 mg of MWCNTs followed by sonication using a Bandelin Sonorex RK100 ultrasonic bath for 90 min in a one-pot procedure. The resulting suspension was then treated following the previous procedure.

Scanning electron microscopic images of buckypaper surfaces were carried out by FEI/Quanta FEG 250 scanning electron microscope (SEM, Hillsboro, OR, USA) with an accelerating voltage of 5 kV. Water contact angles were measured after delivering a 5 µL droplet of water at room temperature onto buckypaper surfaces using a Dataphysics OCA 35 system. The surfaces coverage of hemin was calculated (Equation (1)) by integration of the charge recorded under the one-electron oxidation peak of hemin by cyclic voltammetry at 20 mV s^−1^ for a 1.13 cm^2^ hemin-L_bp_ working electrode.
(1)$$\Gamma =\frac{Q}{\mathrm{nFA}}$$
where Q is the integrated charge, F is the Faraday constant, A is the geometric area of the electrode, and n is the number of electrons transferred.

### 2.3. Hollow Bioelectrode Preparation

A MWCNT commercial buckypaper (C_bp_; 60 gsm, reference number NTL-12218) from Nano-TechLab (Yadkinville, NC, USA) was first cut into individual conductive sheet (Ø = 30 mm) to be used as a support. The circumference of the C_bp_ sheet was coated with carbon paste to create a hollow cylinder with a diameter of Ø = 12 mm dedicated to the trapping of a biocatalyst. An electrical wire was attached to the carbon paste and the electrode was subsequently sealed using a L_bp_ or hemin-L_bp_ sheets (Ø = 30 mm). After the deposition of enzyme powder in the microcavity, the hollow bioelectrodes were left to dry in air for 4 h. Laser intensity image of the electrode cavities were carried out using a 3D Laser Scanning Microscope (Keyence VK-X200; Itasca, IL, USA).

### 2.4. Electrochemical Measurements

Electrochemical measurements were performed using a PGSTAT 100 potentiostat operated by Nova software (Herisau, Switzerland). All bioelectrocatalytic tests were performed in aqueous solution (0.1 M phosphate buffer) at controlled temperature, agitation (500 rpm) and O_2_-saturated condition, using a platinum wire as counter electrode, Ag/AgCl (KCl saturated) as a reference electrode and the hollow electrodes as working electrodes in a conventional three-electrode electrochemical cell. All potentials are given with respect to RHE (E_RHE_ = E_Ag/AgCl_ + 0.197 V).

## 3. Results and Discussion

The hollow electrode was created by assembling a disk (diameter 30 mm) of a commercial buckypaper (C_bp_) with a similar disk of buckypaper (L_bp_) made by filtration of an organic dispersion of multiwalled carbon nanotubes (MWCNTs) by sticking the periphery of the discs with carbon paste. C_bp_ presents a more porous structure than L_bp_ which facilitates the penetration of water and substrates inside the inner cavity. In contrast, L_bp_ exhibits a denser and compact structure composed of significantly smaller nanotubes (diameters 10–20 nm) which allow direct electron transfer (DET) with redox enzymes [9]. The electrical connection of the two glued disks was carried out by insertion of a metallic wire into the carbon paste, the thickness of which defines the volume of the microcavity. Figure 1A shows an image of the resulting hollow electrode as well as a section of the inner cavity that corresponds to about 28 ± 4 µL.

With the aim to illustrate the concept of entrapment and electrical wiring of an enzyme within a hollow bioelectrode, bilirubin oxidase (BOx) was chosen as a model enzyme. BOx is a multicopper oxidase typically employed as the biocatalyst for the four-electron reduction of O_2_ to H_2_O and widely used for the production of biofuel cells [10,11,12,13]. Thus, 2 mg of BOx powder was deposited on one disk before the formation of the cavity (diameter 12 mm). The resulting bioelectrode was then immersed into a in 0.1 mol L^−1^ phosphate buffer (pH 6.5) and the potential electroactivity of BOx towards the reduction of O_2_ was investigated by cyclic voltammetry. Figure 1B shows the cyclic voltammograms recorded at 1 mV s^−1^ under argon and O_2_-saturated conditions. In presence of O_2_, a catalytic current clearly appears with onset potential *ca.* 0.55 V which corresponds to the conventional catalytic phenomena observed for bioelectrodes based on a DET with immobilized BOx [14]. This catalytic current confirms the penetration of water and substrate, the solubilization of the enzyme powder in the inner microcavity and the ability of the hollow electrode to establish an electrical communication with BOx.

As previously reported, the presence of different types of porphyrins adsorbed on MWCNTs coatings or buckypapers generally induces a BOx orientation favorable to a DET between its T1 Cu center and carbon nanotubes and thus enhances the intensity of the catalytic current for O_2_ reduction [14]. The improvement of the electrocatalytic activity of the hollow bioelectrode was therefore studied via the modification of the L_bp_ by an iron-protoporphyrin. For this purpose, hemin was solubilized in the MWCNTs dispersion and firmly adsorbed on MWCNTs walls by strong π–π interactions before the filtration step. The resulting bioelectrode based on hemin-L_bp_ glued with C_bp_ was tested towards O_2_ reduction. As expected, a marked increase in current intensity (+52% at 0.5 V) and an improved onset potential (*ca.* 0.77 V) were recorded corroborating the efficient orientation of BOx induced by the adsorbed hemin (Figure 1B).

In addition to the orientation of the BOx by electrostatic interactions beneficial to the DET, the presence of carboxylic groups on the hemin confers a less hydrophobic character to the MWCNTs and therefore to the buckypaper, thus facilitating its wettability. This hypothesis has been corroborated by contact angle measurements made with the different buckypapers. The water contact angles of L_bp_ and C_bp_ were 128 ± 7° and 131 ± 3°, respectively. In contrast, the contact angle of hemin-L_bp_ could not be measured, confirming that modification of MWCNTs by hemin rendered the surface hydrophilic (Appendix A). Moreover, top-down SEM images of buckypapers (Appendix A) showed a subtle morphologic change of hemin-L_bp_ compared to L_bp_ due to the presence of hemin layers. Furthermore, the amount of immobilized hemin on L_bp_ was estimated via the charge recorded under the Fe^2+/3+^ redox couple of hemin at E_1/2_= −0.140 V vs. RHE. A resulting surface coverage with hemin of 2.66 × 10^−8^ mol cm^−2^ (Appendix A) theoretically corresponds to the formation of 380 compact monolayers on flat surface illustrating the 3D functionalization of L_BP_ [15].

The storage stability of two hollow bioelectrodes maintained in an aqueous solution and containing 2 mg of BOx, was investigated by recording periodically their faradaic catalytic current for O_2_ reduction at 0.5 V as a function of time (Figure 1C). After an initial drastic decrease in catalytic current for approximately 20 days for both bioelectrodes, a near stabilization of the catalytic process occurs, exhibiting a small continuous decrease in catalytic current of 1.9 ± 0.3 and 2.2 ± 0.1 µA cm^−2^ day^−1^ for bioelectrode based on L_bp_ and hemin-L_bp_, respectively. It should be noted that the bioelectrode without hemin loses almost totally its electroactivity after 75 days whereas the bioelectrode based on hemin-L_bp_ retains 30% and 11% of its initial activity after 3 and 6 months respectively.

The manufacturing process of the hollow electrode makes it possible to easily modulate the quantity of enzyme trapped inside the hollow electrode. Thus, the influence of immobilized amount of BOx (0.16–4 mg) on the electrocatalytic properties of the resulting bioelectrodes based on hemin-L_bp_ was studied (Figure 2A). As expected, the catalytic current increases with increasing amounts of BOx up to 1 mg and reaches a plateau for higher amounts. Above 1 mg of trapped enzyme, the electroenzymatic reaction could be limited by the electroactive surface or the diffusion of oxygen inside the cavity. In this context, the experiments were continued with 2 mg of BOx in the microcavity in order to have a reserve of catalyst.

The effect of temperature and pH on the functioning of the trapped enzyme has been studied. The current response of the hollow bioelectrode was measured in the temperature range of 15 to 60 °C. The catalytic current increases to a maximum at 25–37 °C and then decreases sharply, reflecting enzyme deactivation (Figure 2B). Regarding the pH dependence in the range 4–8.5, it appears that the bioelectrocatalytic response is good at pH values ranging from 6 to 7 (83% of activity), the maximum current being recorded at pH 6.0. Taking into account that a strong decrease in activity is observed between pH 5 and 6, the experiments were carried out at pH 6.5 to avoid pH effects (Figure 2C). With the aim to estimate the turnover frequency for BOx in solution within the microcavity, the charge related to the chronoamperometric response of the bioelectrode based on hemin-L_bp_ at 0.5 V for 15 min was recorded (Figure 2D). By the integration of the anodic current area to give the transferred charges from the chronoamperometric measurement and taking into account 4 e^−^ for O_2_ reduction, a TOF of 0.3 s^−1^ was calculated which represents 14% of the specific activity of the enzyme (1.96 U mg^−1^ BOx).

If the electroenzymatic reaction is limited by the need to have contact between enzyme and buckypaper surface (DET), one possibility to improve the catalytic current would be to introduce a freely diffusing redox mediator into the confined solution to electrically connect BOx in solution via mediated electron transfer (MET). As ABTS has already been successfully used for MET with BOx [16], 0.5 mg of ABTS and 2.0 mg of BOx were trapped in the cavity of the hollow electrode based on hemin-L_bp_. Figure 3A shows a clear increase (+40%) in the value of the faradaic catalytic current from −1.35 mA cm^−2^ to −1.9 mA cm^−2^ at 0.5 V, corroborating the electrical wiring of BOx by two phenomena: DET and MET by ABTS.

In addition, the long-term operational stability of the bioelectrode containing BOx and ABTS was studied by chronoamperometry at 0.5 V. A remarkable stability of the cathodic current (loss of 2% after 8 h) seems to indicate the absence of ABTS release (Figure 3B). After 23 h of intermittent operation over 7 days, 218.6 C were recorded corresponding to the reduction of 5.66 × 10^−4^ mol of O_2_ and leading to a turnover number of 19245, the loss of catalytic current being only 54% after 7 days.

## 4. Conclusions

This work reports on the easy elaboration of hollow planar electrodes based on buckypapers and their successful application to the trapping of an enzyme powder in a microcavity. The resulting bioelectrodes are permeable to water but do not allow enzyme permeation. The entrapment of BOx as electrocatalyst model leads to a bioelectrode exhibiting efficient electroenzymatic reduction of O_2_ by direct electron transfer with BOx molecules in solution. The bioelectrocatalysis process was improved using hemin and ABTS for orientation and electrical wiring of BOx, respectively. In addition to long-term operation and storage stability of the hollow bioelectrode, this design allows co-immobilization of enzymes and redox mediators in the microcavity and modulation of the amount of entrapped enzyme. This concept paves the way for the development of a new generation of enzyme electrodes as biosensors or biofuel cells.

## Figures and Tables

**Figure 1 nanomaterials-12-02399-f001:**
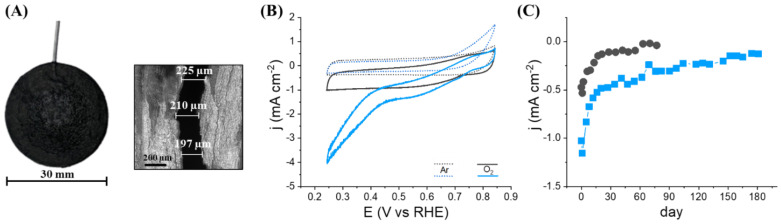
(**A**) Face view photography of the hollow MWCNT-based buckypaper electrode and laser intensity image of vertical cross section of a hollow electrode. (**B**) Cyclic voltammograms recorded in phosphate buffer (pH 6.5; 1 mV s^−1^) at hollow electrodes based on L_bp_ (black) or hemin-L_bp_ (blue) glued with C_bp_ and containing BOx (2.0 mg) trapped inside the electrode cavity. Buffer under argon (dashes) or O_2_ (solid lines). (**C**) Evolution of the faradic catalytic current at 0.5V for the reduction of O_2_ as a function of time recorded periodically on hollow electrodes based on L_bp_ (black) or hemin-L_bp_ (blue) and containing 2 mg of BOx. Both bioelectrodes are stored in 0.1 M phosphate buffer (pH 6.5).

**Figure 2 nanomaterials-12-02399-f002:**
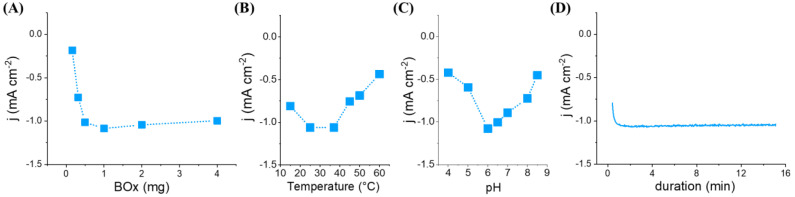
(**A**) Influence of various amounts of BOx entrapped inside the hollow electrode based on hemin-L_bp_ on the electrocatalytic reduction of O_2_. Applied potential 0.5 V vs. RHE. Faradaic catalytic current recorded at 0.5 V on a hollow electrode based on hemin-L_bp_ in oxygen-purged 0.1 M phosphate buffer and containing 2 mg of BOx; (**B**) plot of the current as a function of temperature at pH 6.5; (**C**) Plot of the current as a function of pH at 25 °C. (**D**) Chronoamperometric measurement at 0.5 V in oxygen-purged 0.1 M phosphate buffer (pH 6.5) at 25 °C.

**Figure 3 nanomaterials-12-02399-f003:**
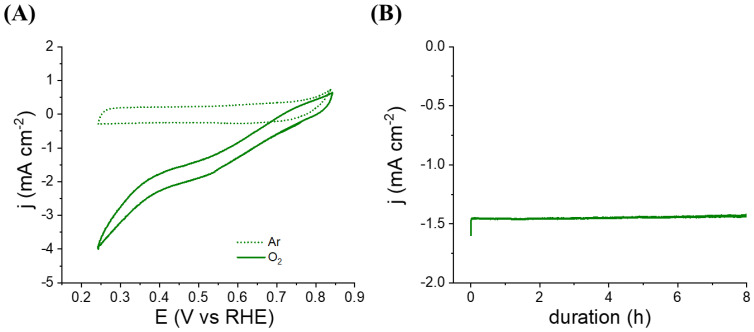
(**A**) Cyclic voltammograms recorded in phosphate buffer (pH 6.5; 1 mV s^−1^) at a hollow electrode based on hemin-L_bp_ containing 0.5 mg of ABTS and 2.0 mg of BOx under argon (dashes) or O_2_ (solid line). (**B**) Chronoamperometric measurement at 0.5 V under oxygen-purged phosphate buffer.

## Data Availability

The data that support the findings of this study are available with the corresponding author, Dr. Serge Cosnier, upon reasonable request.

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
