# Peer review of "Hollow Bioelectrodes Based on Buckypaper Assembly. Application to the Electroenzymatic Reduction of O2"

_nanomaterials, 2022, doi:10.3390/nano12142399_

Round 1
Reviewer 1 Report
This is a nice work. The hollow bioelectrode using buckypaper assembly was constructed and the stability was improved a lot. The work was well-structured and the performance of platform was investigated using DET and MET with BOD as a model. It can be accepted after demonstrating the following comments:
1) Scheme should be given to clearly demonstrate the fabrication of the electrode.
2) Some small mistakes should be corrected for example in the abstract” 5.66 10-4 mol of O2”
3) Why choose the commercial bulkypaper as carrier, what about the performance of buckeypaper made in the lab as the support.
4) The figures should be polished, such as the Figure 2.
5) How to control the volume of the microcavity and the size?
6) Give the explanation on the drastic decrease in the catalytic current during the 20 days!
7) The details information of calculation of TOF should be given in the present work!
Author Response
Grenoble, July 7, 2022
Dr. Winnie Tong
Assistant Editor
Manuscript ID: nanomaterials-1795881
Title: Hollow Bioelectrodes based on Buckypaper Assembly. Application to the Electroenzymatic Reduction of O2.
Dear Dr. Tong,
We appreciate the time and effort spent by yourself and the reviewers on reviewing this manuscript. We understand that our manuscript may have lacked clarity. We have therefore completed the revision of our manuscript and the electronic supplementary material (the changes were highlighted in yellow) in conformity with the suggestions of the reviewers. The corrections and detailed explanations are listed below point by point.
I hope that these modifications will satisfy the points raised by the reviewers and clarify the ideas and results that we intend to publish in Nanomaterials.
Comments of the Reviewer 1:
This is a nice work. The hollow bioelectrode using buckypaper assembly was constructed and the stability was improved a lot. The work was well-structured and the performance of platform was investigated using DET and MET with BOD as a model. It can be accepted after demonstrating the following comments:
- Scheme should be given to clearly demonstrate the fabrication of the electrode.
We thank the reviewer for his kind suggestions. An illustrative representation of the hollow electrode composition will be represented in the Graphical Abstract.
- Some small mistakes should be corrected for example in the abstract” 5.66 10-4 mol of O2”
We appreciate the referee's observation. The manuscript has been carefully checked.
- Why choose the commercial bulkypaper as carrier, what about the performance of buckeypaper made in the lab as the support.
The hollow electrode was created using on one side a commercial buckypaper which has a greater macroporosity than the bukypaper manufactured by our laboratory (see for example the reference 9) in order to facilitate the penetration of water and substrates inside the inner cavity. Nevertheless, we are currently studying some hollow electrodes made only with lab-made buckypapers.
- The figures should be polished, such as the Figure 2.
We fully agree with the reviewer's suggestion. The figures were revised and modified.
- How to control the volume of the microcavity and the size?
We thank the Reviewer #1 for allowing us to clear up this point. Based in a simple assembly of two buckypapers, the hollow electrode dimensions and the cavities geometric shapes may be chosen according to the research purpose. In this work was used a temporary cylindrical template of Teflon (Ø = 12 mm) to create a hollow cylinder dedicated to the trapping of a biocatalyst, while the controlled height of the carbon paste was used to define volume of cavity.
- Give the explanation on the drastic decrease in the catalytic current during the 20 days!
The Reviewer #1 raised an important point here in which is not yet completely understood. It is envisaged that during the first 20 days, enzymes adsorb on the initially virgin surface of the buckypapers in a configuration that does not allow electronic exchanges, even are denatured in the adsorbed state. This phenomenon must lead to a reduction in the electroactive surface and on the other hand a reduction in the permeability of the buckypaper.
- The details information of calculation of TOF should be given in the present work!
We followed the reviewer recommendation, and this point was briefly added in the text page 5, lines 227-230.
Reviewer 2 Report
The authors assembled a bioelectrode for enzymatic electroreduction of oxygen in the phosphate buffers. However, the lack of materials characterization before and after electrochemistry, the O2 reduction performance is low and not studied accordingly. I understand that the bioelectrode are not comparable to commercial materials and the gap is wide, but still the materials reported here are neither efficient nor stable from my view. Please find my comments here:
1. Please convert potential to RHE and mention the medium in which the experiments were performed in the abstract as well.
2. In the abstract the authors mentioned the storage stability although they do conversion, why?
3. What is the relation between storage and stability here, again you mentioned in the introduction? The current density retention here should be discussed carefully, not storage!
4. It would be better to convert current/mA into current density/mAcm-2 throughout the ms and in figures as well.
5. CV is not enough to evaluate your materials ORR performance, LSV at different rotations is required as well to study kinetics and determine number of electrons transfer.
6. What is R2 in figure S2, you should explain.
7. It is clear your materials lose its activity after around 120 days, which it should be considered as O2 reduction catalyst for biofuel cells?
8. In my opinion studying the performances of your catalysts widely at room temp widely is more important than tuning temp and pH.
9. What is the benefit from doing chronoamperometry test for 15 min? it means nothing at all!
10. Why did you consider 4e- for O2 reduction?
11. Figure 3B is nice
Author Response
Department of Molecular Chemistry UMR CNRS 5250
Dr. Serge Cosnier
Director of Research at CNRS
E-mail : Serge.Cosnier@univ-grenoble-alpes.fr
Département de Chimie Moléculaire UMR CNRS 5250
570 rue de la Chimie, Bâtiment Nanobio
Université Grenoble-Alpes, CS 40700
38058 Grenoble cedex 9, France
Grenoble, July 7, 2022
Manuscript ID: nanomaterials-1795881
Title: Hollow Bioelectrodes based on Buckypaper Assembly. Application to the Electroenzymatic Reduction of O2.
We appreciate the time and effort spent by yourself and the reviewers on reviewing this manuscript. We understand that our manuscript may have lacked clarity. We have therefore completed the revision of our manuscript and the electronic supplementary material (the changes were highlighted in yellow) in conformity with the suggestions of the reviewers. The corrections and detailed explanations are listed below point by point.
I hope that these modifications will satisfy the points raised by the reviewers and clarify the ideas and results that we intend to publish in Nanomaterials.
Comments of the Reviewer 1:
This is a nice work. The hollow bioelectrode using buckypaper assembly was constructed and the stability was improved a lot. The work was well-structured and the performance of platform was investigated using DET and MET with BOD as a model. It can be accepted after demonstrating the following comments:
- Scheme should be given to clearly demonstrate the fabrication of the electrode.
We thank the reviewer for his kind suggestions. An illustrative representation of the hollow electrode composition will be represented in the Graphical Abstract.
- Some small mistakes should be corrected for example in the abstract” 5.66 10-4 mol of O2”
We appreciate the referee's observation. The manuscript has been carefully checked.
- Why choose the commercial bulkypaper as carrier, what about the performance of buckeypaper made in the lab as the support.
The hollow electrode was created using on one side a commercial buckypaper which has a greater macroporosity than the bukypaper manufactured by our laboratory (see for example the reference 9) in order to facilitate the penetration of water and substrates inside the inner cavity. Nevertheless, we are currently studying some hollow electrodes made only with lab-made buckypapers.
- The figures should be polished, such as the Figure 2.
We fully agree with the reviewer's suggestion. The figures were revised and modified.
- How to control the volume of the microcavity and the size?
We thank the Reviewer #1 for allowing us to clear up this point. Based in a simple assembly of two buckypapers, the hollow electrode dimensions and the cavities geometric shapes may be chosen according to the research purpose. In this work was used a temporary cylindrical template of Teflon (Ø = 12 mm) to create a hollow cylinder dedicated to the trapping of a biocatalyst, while the controlled height of the carbon paste was used to define volume of cavity.
- Give the explanation on the drastic decrease in the catalytic current during the 20 days!
The Reviewer #1 raised an important point here in which is not yet completely understood. It is envisaged that during the first 20 days, enzymes adsorb on the initially virgin surface of the buckypapers in a configuration that does not allow electronic exchanges, even are denatured in the adsorbed state. This phenomenon must lead to a reduction in the electroactive surface and on the other hand a reduction in the permeability of the buckypaper.
- The details information of calculation of TOF should be given in the present work!
We followed the reviewer recommendation, and this point was briefly added in the text page 5, lines 227-230.
Comments of the Reviewer 2:
The authors assembled a bioelectrode for enzymatic electroreduction of oxygen in the phosphate buffers. However, the lack of materials characterization before and after electrochemistry, the O2 reduction performance is low and not studied accordingly. I understand that the bioelectrode are not comparable to commercial materials and the gap is wide, but still the materials reported here are neither efficient nor stable from my view. Please find my comments here:
The weak point of enzyme electrodes is almost always the biological element. The enzyme generally immobilized on the electrode, is denatured over time, and constitutes the limiting factor for the electrocatalytic activity of the bioelectrode. We have chosen the enzymatic oxygen reduction as a model reaction to show the possibility of trapping and connecting an enzyme in a microvolume while retaining electroenzymatic activity over a long period of time. Our goal was not to study in depth this reaction which is widely described via the numerous publications of bilirubin oxidase electrodes but rather to illustrate a new concept of enzyme electrode manufacturing.
- Please convert potential to RHE and mention the medium in which the experiments were performed in the abstract as well.
In agreement with the reviewer request, we have modified the reference for the potential values and added the composition of the aqueous solution.
- In the abstract the authors mentioned the storage stability although they do conversion, why?
We thank the Reviewer #2 for allowing us to clear up this point. Here, what should be understood is that the hollow bioelectrodes are stored in between experiments in phosphate buffer solution. The cyclic voltammetry characterizations illustrate the electrocatalytic current and therefore the residual activity of the trapped enzyme. This evaluation of the activity is related to the storage time in the aqueous solution and thus indicates the evolution of the bioelectrode over a long period (up to 6 months). This is very important because in general the enzymes are deactivated fairly quickly and on the other hand it illustrates the tightness of the microcavity with respect to the release of the enzymes emphasizing the very slight decrease in the electroenzymatic activity.
- What is the relation between storage and stability here, again you mentioned in the introduction? The current density retention here should be discussed carefully, not storage!
We appreciate the referee's observation. In order to eliminate any misunderstanding, storage and stability are linked. Storage is only related to the fact that the biofuel cells are stored before and in between experiments in buffer solution. That leads to a leaching out of the enzyme and a direct loss in performances enhanced by the deactivation of the immobilized enzyme. This last point refers then to what we call stability which is directly related to the loss in performances of the biofuel cell over time.
- It would be better to convert current/mA into current density/mAcm-2 throughout the ms and in figures as well.
In accordance with the reviewer's suggestion, we have added the current density values throughout the manuscript.
- CV is not enough to evaluate your materials ORR performance, LSV at different rotations is required as well to study kinetics and determine number of electrons transfer.
Thank you for this point. We realize that the LSV at different rotation is important to learn more about our system, but unfortunately our concept of electrode does not allow us to carry out experiments of this type. To achieve this, it would be necessary to find the means for connecting our electrode to a rotating system. This is a good point to reflect on for the future works.
- What is R2 in figure S2, you should explain.
We thank the Reviewer #2 remark. The R2 mentioned in the Figure S2 corresponds to the coefficient of determination. This point was added in the Figure S2 caption.
- It is clear your materials lose its activity after around 120 days, which it should be considered as O2 reduction catalyst for biofuel cells?
The use of biofuel cells can only consist in generating electricity for a short period (a few hours or even a few days) for single-use sensor applications (see for example the startup BeFC which markets this type of biofuel https://www.befc.global/). In this case, a lifetime of 120 days is a great advantage.
- In my opinion studying the performances of your catalysts widely at room temp widely is more important than tuning temp and pH.
We agree with the referee that studies of our catalysts at room temperature are highly important, especially for their future use in vitro testing. Our studies on the influence of temperature and pH on the electroactivity of the bioelectrode were mainly aimed at showing that trapping in the hollow electrode did not modify the characteristics of the enzyme with respect to these parameters. In addition, these studies have enabled us to define the optimal conditions for our biocathodes. In addition, future prospects are to implant our bioelectrodes in vivo, hence the need to know the influence of physiological temperature and pH (37°C and pH 7.4).
- What is the benefit from doing chronoamperometry test for 15 min? it means nothing at all!
Our objective was to know the initial enzymatic activity by measuring the catalytic current. We therefore measured this parameter for 15 min to be sure to have a stable value. The advantage is that by construction, the quantity of enzyme immobilized in the microcavity is known, contrary to the majority of bioelectrodes made up to now. It is therefore possible to estimate an initial specific activity of the enzyme and to compare it with the conventional value in solution.
- Why did you consider 4e- for O2 reduction?
We thank the Reviewer #2 for this question. In our work we did not determine in practice the number of transfer electrons as the reviewer suggested to us previously (see question 5). We based ourselves on the theory and the studies that already exist in this field.
To achieve this, in our biocathodes we use Bilirubin oxidase as an enzyme. Bilirubin oxidase is a Multicopper oxidase (MCO) subclass enzyme that was discovered in 1981 by Tanaka and Murao and is very used now days for the oxygen reduction. In 1996 Solomon and all showed the molecular mechanism for the four - electron reduction of O2 to H2O via spectroscopic studies combined with crystallography. In 2007 Sakurai and Kataoka also showed than BOx reduced O2 by 4-electron transfer. Brocato and all show in their work that the mechanism of oxygen reduction by bilirubin oxidase involves 4-electron transfer following the reaction: O2 + 4H+ + 4 e = 2H2O. They did studies using an electrode rotating with BOx at different speeds and they monitored the production of H2O2. They demonstrated through their work that the mechanism of O2 reduction by bilirubin oxidase is 4-electron because there was no peroxide intermediate detected at all (Brocato and all, 2012). In fact, this type of enzyme contains four Cu active sites: one Cu ion at T1 and three Cu ions in tri-nuclear cluster (T2/T3). The single-electron oxidation of substrates takes place at the proximal T1 site, with the 4-electron reduction of O2 taking place at the tri-nuclear cluster. Following the oxidation of substrates, 4 electrons are quickly transferred between the T1 site and the TNC (in 4*1 electron transfers) of MCOs via a His-Cys-His chain (Milton R.D and all, 2013).
There are many other studies on this subject, that’s why in this work we decided to consider 4e for reduction of O2. The references cited above have been added to the manuscript.
- Figure 3B is nice
Thanks for this kind remark
I hope that the aforementioned comments and modifications will help to clarify the ideas we intend to publish in Nanomaterials.
Sincerely yours,
Dr. Serge Cosnier

Round 2
Reviewer 2 Report
I am very satisfied with the authors response and have no more comments on this ms.